# Glomerular Hyperfiltration: A Marker of Fibrosis Severity in Metabolic Associated Steatotic Liver Disease in an Adult Population

**DOI:** 10.3390/ijms242115837

**Published:** 2023-10-31

**Authors:** Andrea Dalbeni, Marta Garbin, Mirko Zoncapè, Sara Romeo, Filippo Cattazzo, Anna Mantovani, Annalisa Cespiati, Anna Ludovica Fracanzani, Emmanouil Tsochatzis, David Sacerdoti, Alessandro Mantovani, Rosa Lombardi

**Affiliations:** 1General Medicine C, Department of Medicine, University and Azienda Ospedaliera Universitaria Integrata of Verona, 37126 Verona, Italy; andrea.dalbeni@aovr.veneto.it (A.D.); martagarbin94@hotmail.it (M.G.); mirko.zoncape@univr.it (M.Z.); sara.romeo@studenti.univr.it (S.R.); cattazzo.f@gmail.com (F.C.); anna.mantovani@aovr.veneto.it (A.M.); 2Liver Unit, Department of Medicine, University and Azienda Ospedaliera Universitaria Integrata of Verona, 37126 Verona, Italy; david.sacerdoti@univr.it; 3UCL Institute for Liver and Digestive Health, Royal Free Hospital and UCL, London NW3 2PF, UK; e.tsochatzis@ucl.ac.uk (E.T.); rosa.lombardi@unimi.it (R.L.); 4SC Medicina ad Indirizzo Metabolico, Fondazione IRCCS Ca’ Granda Ospedale Maggiore Policlinico di Milano, Department of Pathophysiology and Transplantation, University of Milan, 20122 Milan, Italy; anna.fracanzani@unimi.it; 5Endocrinology Unit, Department of Medicine, University and Azienda Ospedaliera Universitaria Integrata of Verona, 37126 Verona, Italy; alessandro.mantovani@univr.it

**Keywords:** MASLD, hepatic fibrosis, FibroScan, kidney disease, glomerular filtration rate, steatotic liver disease

## Abstract

Glomerular hyperfiltration (GH) is an increase in the glomerular filtration rate, possibly progressing to chronic kidney disease (CKD). Metabolic-associated steatotic liver disease (MASLD) is linked to an increased risk of CKD, especially if fibrosis is present; however, the association between GH and MASLD has not been explored. To evaluate GH prevalence in MASLD and its possible correlation with liver fibrosis. 772 consecutive patients with ultrasound MASLD (mean age 47.3 ± 8.9 years, 67.1% males) were enrolled. GH was defined as estimated glomerular filtration rate (eGFR) greater than the upper quartile of values in the cohort. Liver stiffness measurement (LSM) by FibroScan ≥ 7.2 kPa suggested liver fibrosis. GH was present in 20% of patients, liver fibrosis in 30%. In total, 53.4% of the cohort was obese, 40.9% hypertensive, 36.3% diabetic and 70.8% dyslipidaemic. GH patients compared to non-GH were significantly younger (38.4 ± 8.3 vs. 49.5 ± 7.7, *p* < 0.001), with higher prevalence of LSM > 7.2 kPa (35.5% vs. 29%, *p* < 0.001), without any difference in metabolic comorbidities. In multivariate analysis, age (OR 0.85, CI 95% 0.82–0.87) and significant fibrosis (OR 1.83; CI 95%1.10–3.03) remained independently associated with GH, regardless of the presence of metabolic alterations and nephrotoxic drugs. GH, an early marker of renal damage, is highly prevalent in MASLD and is associated with hepatic fibrosis. GH may be considered an early marker of both liver and renal disease and its recognition could prompt the management of risk factors aimed at preventing the progression of both hepatic and renal disease.

## 1. Introduction

Glomerular hyperfiltration (GH) can be defined as a supraphysiological elevation in the glomerular filtration rate (GFR) that occurs when the single-nephron glomerular filtration rate (SN-GFR) increases in all or some kidney nephrons. Persistent increases in the SN-GFR can eventually lead to proteinuria, glomerulosclerosis and decline in kidney function [1]. Therefore, GH is considered the early phase of the pathogenetic process that leads to chronic kidney disease (CKD), which is conversely defined by a reduced GFR (<60 mL/min/1.73 m [2]) and/or markers of kidney damage, indicated by moderately increased albuminuria (albumin-to-creatinine ratio (UACR) 30–300 mg/g), severely increased UACR (>300 mg/g) or nephrotic-range proteinuria (UACR > 2200 mg/g). It is known that CKD patients, at any stage of the disease, are strongly predisposed to develop cardiovascular disease (CVD) and CKD is a well-known risk factor for all-cause and cardiovascular mortality [2,3]. Therefore, finding early markers of renal disease, as well as novel prognostic factors associated with its development and progression becomes crucial. Despite the fact that clinical implications of GH remain unclear, it has been associated with increased cardiovascular mortality and all-cause mortality in an apparently healthy adult population, regardless of the smoking status [3]. Indeed, GH is significantly associated with insulin resistance, prediabetes and type 2 diabetes (T2DM), prehypertension and hypertension and obesity [3], all well-known cardiovascular risk factors.

In addition, GH has also been associated with other pathological conditions, such as dementia [4], sleep disturbances [5] and cancers, the latter both in the adult and infant population [6,7].

The metabolic alterations related to GH also predispose to the accumulation of fat in the liver, configuring the condition of hepatic steatosis [8]. In June 2023, an international consensus panel introduced the term steatotic liver disease (SLD) as an umbrella term encompassing the various aetiologies of hepatic steatosis, and particularly the concept of metabolic-associated steatotic liver disease (MASLD), formerly known as non-alcoholic fatty liver disease (NAFLD), which is defined by the presence of hepatic steatosis with at least one cardiometabolic risk factor, in the absence of excessive alcohol intake or other known causes of liver disease [9].

MASLD includes several stages of liver disease, possibly progressing towards metabolic-associated steatohepatitis (MASH), cirrhosis and end-stage liver disease [2,10]. Furthermore, hepatic steatosis exposes patients to higher risk of CVD [2,10,11], as well as an increased risk of developing CKD, and the more the advanced stages of liver disease the more the severity of renal impairment [10,12].

As GH and MASLD share the same cardiovascular and metabolic risk factors, a possible association between these two conditions could have been expected. Nevertheless, data in literature on this topic focus on the previous definition of NAFLD, are scarce and include heterogeneous and small cohorts, usually using non-invasive scores for the diagnosis of hepatic steatosis.

Therefore, the aim of our study is to evaluate the prevalence of GH and its associated factors in a wide multicentric adult population of ultrasound-proven MASLD subjects, as well as a possible positive correlation between GH and advanced forms of liver disease.

## 2. Results

### 2.1. Baseline Characteristics of the Whole Cohort of MASLD Patients

Out of 1497 MASLD patients evaluated, 772 subjects were enrolled according to the selection criteria (Figure 1).

The general characteristics of the whole population are described in Table 1. The mean age was 47.3 ± 8.9, with 67.1% males. In total, 15.8% of the cohort were current smokers. GH (i.e., GFR > 110 mL/min) was present in 152 (19.6%) patients, with mean GFR of 96.6 ± 14.8 mL/min, according to the CKD-EPI formula. As for metabolic comorbidities, more than half of the cohort was obese, 40.9% presented hypertension, 36.3% T2DM and 70.8% dyslipidaemia. As for liver disease, 29.5% of the cohort presented a severe grade of steatosis at US and the mean CAP values were 316 ± 56 dB/m, while presence of liver fibrosis by LSM (i.e., > 7.2 kPa) was present in 234 (30.3%) patients, with a median value of 7.6 kPa (IQR 2.2–52.2).

### 2.2. Baseline Characteristics of the Whole Cohort of MASLD Patients According to the Presence vs. Absence of Glomerular Hyperfiltration (GH)

As reported in Table 1, comparing patients with and without hyperfiltration, the first group was significantly younger compared to the other (38.4 ± 8.3 vs. 49.5 ± 7.7, *p* < 0.001), whereas no difference was observed concerning smoking status or ethnicity. Similarly, prevalence of all metabolic comorbidities, except for presence of hypertension (higher in non-hyperfiltrating subjects), was superimposable between the two groups, as well as use of any antihypertensive drugs, including angiotensin-converting enzyme inhibitors (ACEi) or angiotensin receptor blockers (ARBs), diuretics or statins.

As for liver disease, a significantly higher prevalence of severe steatosis (39.5% vs. 27.1%, *p* = 0.007), as well as increased LSM values (8.0 ± 6.1 vs. 7.5 ± 6.1 kPa, *p* = 0.023) were observed in the GH group compared to its counterpart. Similarly, a higher prevalence of fibrosis by LSM > 7.2 kPa was present in hyperfiltrating patients compared to those who did not (35.5% vs. 29%, *p* < 0.001).

Some factors were independently associated with the presence of hyperfiltration in the whole cohort of MASLD patients.

As reported in Table 2, in multivariate analysis (adjusted for age, sex, T2DM, obesity, hypertension), age (OR 0.84; 95% CI 0.82–0.87) and LSM by FibroScan (expressed as Log_10_) (OR 6.6, CI 95% 2.2–19.9) were independently associated with the presence of GH, whereas no association of other metabolic features was found. Interestingly, the independent association with LSM was consistent even after adjusting for ACEi/ARBs or diuretic therapy.

When considering the presence of liver fibrosis by FibroScan (i.e., LSM > 7.2 kPa) instead of LSM expressed as linear variable, fibrosis remained independently associated with the presence of GH (Table 3) (OR 1.83, 95% CI 1.10–3.03).

## 3. Discussion

In this multicentric observational study, we have shown in a wide cohort of young adult patients with MASLD that hepatic fibrosis assessed by FibroScan^®^ is significantly associated with GH, an early marker of renal dysfunction.

The association between MASLD, previously defined as NAFLD, and CKD is well established, as both conditions share common risk factors and pathogenetic pathways, mainly insulin resistance, lipid alterations, oxidative stress and inflammation [10]. In fact, as reported in literature, patients with MASLD have an increased risk of prevalent (OR 2.12, 95% CI 1.69–2.66) and incident (HR 1.79, 95% CI 1.65–1.95) CKD, and the greater the severity of liver disease the higher the risk of developing renal dysfunction [6,13,14,15]. In turn, patients with CKD have an increased risk of developing MASLD and fibrosis [12]. Conversely, less is known about the association between hepatic steatosis, and especially fibrosis, and the early stages of renal damage. GH is considered an early marker of CKD and it has been associated with dysmetabolism as insulin resistance, T2DM, hypertension and obesity [3], known risk factors for both MASLD and CVD. In fact, hyperfiltration seems to be sustained by increased blood pressure, glucose load and metabolic dysfunction and it may enhance albumin ultrafiltration and excretion, thus causing renal stress and ultimately leading to kidney failure [16,17]. In addition, GH predisposes to increased risk of fatal and non-fatal cardiovascular events [3,18]. This is of crucial interest given that MASLD exposes patients to increased risk of cardiovascular complications and mortality, mainly driven by the presence of hepatic fibrosis [19,20].

In our study, including more than 700 Caucasian subjects with MASLD, hepatic fibrosis detected by FibroScan^®^ was associated with GH. Most importantly, this association was independent of the presence of metabolic alterations and age, all known risk factors for the progression of both hepatic and renal disease. Similarly, the association was not impaired by the use of potentially nephrotoxic drugs. Therefore, the detection of both hepatic fibrosis and renal dysfunction in the early stages could be of pivotal importance in order to manage predisposing factors and prevent progression of both conditions to unfavourable outcomes. Our results are in line with what is reported in the literature, even though data are scarce and mainly focused on the association between GH and hepatic steatosis, at that time defined as NAFLD. In fact, GH has been previously associated with the presence of hepatic steatosis in 154 Spanish subjects with metabolic syndrome, with a fivefold increased risk of having steatosis compared to participants not hyperfiltrating [2]. However, this study was monocentric, included a small cohort and did not focus on hepatic fibrosis. Another retrospective study including a paediatric population of 179 obese children with biopsy-proven steatosis, demonstrated that hyperfiltration present in the 20% of the sample (arbitrarily diagnosed by GFR > 130 mL/min and chosen as the average of other studies in literature) was an independent risk factor for the histological severity of liver disease defined by the NAFLD activity score > 5 (OR, 2.96; 95% CI, 1.49–5.87) [21]. Conversely to our results, no association was found with hepatic fibrosis, but the different population studied and the low prevalence of fibrosis in the paediatric population (around 8%) may explain this difference. Another recent study conducted on a large cohort of Spanish patients with pre-diabetes and visceral adiposity showed an independent association between liver steatosis and GH, and the age-mediated GFR reduction appears to be potentiated by MASLD [22]. Finally, a study conducted In an Asiatic population of 147,162 Korean subjects without CKD nor hepatic steatosis and followed up for 4 years showed that the presence of GH at baseline (present in 5% of the cohort) was an independent risk factor for the development of liver fat diagnosed by US (OR 1.21, CI 95% 1.14–1.29) and for its progression to fibrosis assessed by the NAFLD fibrosis score (OR 1.42, CI 95% 1.11–1.82) [23]. In addition, the persistence of GH over time further increased this risk up to 13%. Our study is consistent with the Asian study, not only in confirming the association between GH and fibrosis risk, but it is the first one reporting an independent association between the presence and severity of liver fibrosis and GH in a large Caucasian population of adult subjects. Furthermore, in our series, fibrosis was assessed by FibroScan^®^, which is a more validated non-invasive tool compared to non-invasive tests [19]. Interestingly, Setti et al., evaluating 1552 non-smoking, non-diabetic Finnish subjects (55% with steatosis diagnosed by the fatty liver index and 5% with GH) followed prospectively for 29 years, found a combined effect of GH and fatty liver on the risk of all-cause mortality and cardiovascular mortality over time compared to each condition alone, supporting the evidence of a synergistic effects of renal and liver dysfunction on long-term adverse outcomes [24].

Our study has some limitations. First of all, the retrospective design does not allow a final causal relationship between liver disease and renal impairment assessed by an increased GFR to be established; nevertheless, the association between hepatic steatosis and incident renal dysfunction has been previously demonstrated in prospective studies [13]. Secondly, GFR has not been directly calculated by plasma and urine creatinine levels but using the CKD-EPI formula; however, this equation has been validated either in subjects with normal or deranged renal function [25]. In addition, given the lack of established cut-off to define GH, in this study we have considered the upper quartile of GH values of our cohort, in line with the literature. Finally, we do not have histological data on hepatic fibrosis diagnosed by FibroScan^®^, which, however, has been proposed as a surrogate of hepatic fibrosis by international guidelines [26].

Nevertheless, the strength of our study is to have assessed the simultaneous presence of liver damage and early renal impairment non-invasively in a large series of consecutively enrolled patients with MASLD, targeting a quite asymptomatic population. In addition, the multicentric design of the study, involving both an Italian and a UK cohort, may allow the generalization of our results to the Caucasian population.

## 4. Materials and Methods

This is a three-centres cross-sectional study conducted from January 2014 to June 2022.

All consecutive patients referred to the outpatient hepatology clinics of the General Medicine and Liver Unit of the University Hospital of Verona (Verona, Italy), the Metabolic and Liver Disease Centre of the Policlinico Hospital of Milan (Milan, Italy) and the Royal Free London NHS Foundation Trust, Sheila Sherlock Liver Centre (London, UK) were enrolled according to inclusion and exclusion criteria.

Inclusion criteria were age >18 years and ultrasound (US)-proven MASLD. Exclusion criteria were GFR < 60 mL/min, age > 65 years (we arbitrarily chose this age threshold for possible confounders linked to a high burden of comorbidities in elderly population), other causes of liver disease except MASLD (viral or autoimmune hepatitis, alcohol consumption > 20 g/day or 30 g/day for women and men, respectively, genetic hemochromatosis, Wilson’s disease, a1-antitrypsin deficiency), use of steatosis-inducing drugs and decompensated cirrhosis. Similarly, patients with congestive heart failure, free abdominal fluid or pregnancy were also excluded, according to FibroScan^®^ technical limitations.

The study protocol was approved by the Institutional Ethics Committee of Verona and Milan (Italy) (2524CESC for Verona and N-0006778-U for Milan), while in London (UK) it was a retrospective evaluation as part of an audit. All patients provided written informed consent to participate to the study according to the ethical guidelines of the 1975 Declaration of Helsinki.

### 4.1. Clinical and Biochemical Assessment

At enrolment, anthropometric measurement, medical history, smoking habits and use of current therapy (including antihypertensive agents and statins) were recorded for each participant.

In particular, a body mass index (BMI) of 25–29 and ≥30 kg/m^2^ defined the presence of overweight and obesity, respectively. Hypertension was diagnosed by blood pressure ≥140/90 mmHg or use of any antihypertensive drug and dyslipidaemia by low-density lipoprotein (LDL)-cholesterol > 100 mg/dL, triglycerides > 150 mg/dL and/or high-density lipoprotein (HDL)-cholesterol < 40 mg/dL for men and 50 mg/dL for women, or use of lipid-lowering drugs. Type 2 diabetes was diagnosed by fasting glucose > 126 mg/dL in more than two consecutive measurements or a single value of >200 mg/dL or a glycated haemoglobin (HbA1c) > 6.5% (48 mmol/mol), as well as therapy with insulin or oral/injective hypoglycaemic agents [27,28,29]. In addition, impaired fasting glucose was defined by fasting plasmatic glucose levels between 100 and 126 mg/dL [29].

A complete biochemical panel was assessed and transaminases > 39 U/L and 41 U/L (AST and ALT, respectively), and GGT > 61 UI for men and 36 UI/l for women were considered increased, according to local laboratories cut-offs.

### 4.2. Renal Function Assessment

The estimated glomerular filtration rate (eGFR) was calculated using the new Chronic Kidney Disease Epidemiology Collaboration (CKD-EPI) equation, developed in 2021 [25,30]. The estimated renal function using the CKD-EPI equation was normalised for body surface area (BSA) and expressed as GFR mL/min/1.73 m^2^, correct for age and sex [142 * (serum creatinine/A)^B^ * 0.9938^age^ * (1.012 if female); in male A = 0.9, B = −0.302 if serum creatinine ≤ 0.9, −1.2 if serum creatinine > 0.9; in female A = 0.7, B = −0.241 if serum creatinine ≤ 0.9, −1.2 if serum creatinine > 0.9]. The equation has been validated in populations with normal as well as low GFR [27]. Accordingly, eGFR was converted to absolute values (mL/min) by using the following formula (eGFR mL/min/1.73 m^2^ * BSA)/1.73 m^2^ [31]. BSA was calculated using the DuBois and DuBois equation [32].

Glomerular hyperfiltration was defined as eGFR ≥ the upper quartile of our population, as reported in previous studies [33].

### 4.3. Abdominal Ultrasound

Abdominal US was performed at enrolment by three (one for each centre) experienced sonographers using a 3.5 MHz convex-array probe. Hepatic steatosis was classified as absent, mild, moderate or severe according to the following accepted criteria: hepatorenal echo contrast, liver brightness, deep attenuation and vascular blurring [34,35].

### 4.4. Transient Elastography and Non-Invasive Fibrosis Score

Liver stiffness measurement was assessed using transient elastography (FibroScan^®^, Echosens, Paris, France), using the XL probe for indeterminate measurements with the M one. Results were expressed as liver stiffness measurement (LSM) in kPa and were considered valid if the interquartile range did not exceed 30% of the median value [36]. Given the middle age of our population and GH being an early marker of renal dysfunction, we decided to evaluate the presence of initial stages of hepatic fibrosis by detecting significant liver fibrosis by using a LSM ≥ 7.2 kPa according to international consensus [37].

During FibroScan^®^ assessment, the controlled attenuation parameter (CAP), which is used to quantify hepatic steatosis and is expressed as dB/min, was obtained simultaneously to LSM.

### 4.5. Statistical Analysis

Continuous variables are presented as mean ± standard deviation or median (interquartile range) based on data distribution. Categorical variables are expressed as percentages. Variables with non-normal distribution were transformed in Log_10_. Categorical variables were compared using the Chi-square test. Logistic multivariate regression analyses were performed to determine if any anamnestic or clinical variables could be independently associated with hyperfiltration (odd ratio (OR), with a confidence interval (CI) of 95%). The variable selection was performed through sequential replacement (a stepwise method), which consists of a combination of backward and forward techniques. If the *p*-value was less than 0.05 or above 0.1, the covariates were, respectively, included and excluded from the regression model. Jamovi 2.2.5 solid version was used for all data analyses. All tests were 2-sided, and *p*-values < 0.05 were considered statistically significant.

## 5. Conclusions

In conclusion, this is the first study demonstrating the relationship between hepatic fibrosis and renal dysfunction in precocious stages defined by GH in a wide cohort of MASLD subjects. We have further demonstrated that hepatic fibrosis, diagnosed by FibroScan^®^, was independently associated with GH, a marker of early renal damage. This would open a new prospective in the diagnostic scenario of patients with MASLD. In fact, if, on the one hand, indication to screen MASLD patients for CKD, defined by a decline in GFR, is established, on the other hand, no attention has been paid to date to hyperfiltration, which often precedes CKD. Therefore, screening MASLD subjects for abnormally elevated GFR values could foster an earlier recognition of a possible risk of CKD development. On the other hand, detection of increased GFR in outpatient clinics, may allow the prompt referral of patients to a hepatology setting, prompting actuating intervention strategies able to prevent the occurrence of advanced hepatic and renal damage. This could have an impact either on health or economic aspects given the high burden of cardiovascular morbidity and mortality driven by both MASLD and CKD. Given the retrospective nature of this study, further prospective studies are necessary to validate our findings.

## Figures and Tables

**Figure 1 ijms-24-15837-f001:**
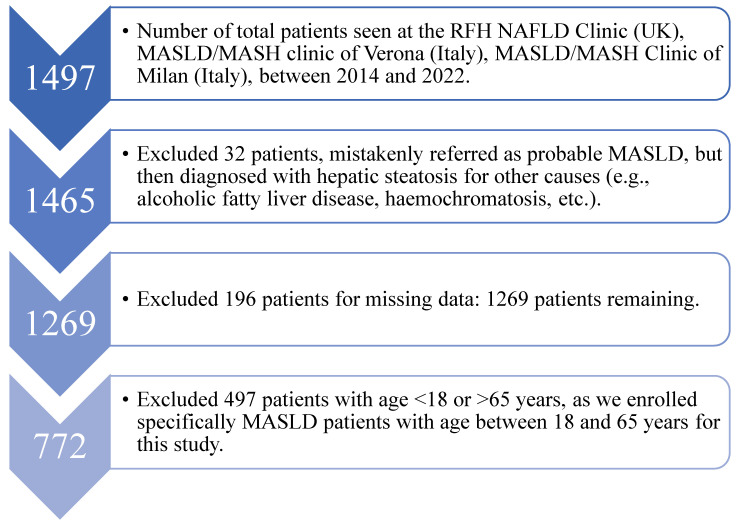
Diagram showing how the patients considered for the present study were selected.

**Table 1 ijms-24-15837-t001:** Descriptive statistics for MASLD population between 18 and 65 years and for the 2 sub-cohorts of hyperfiltrating and non-hyperfiltrating patients.

Variables	Total Population (N = 772)	Non-Hyperfiltrating Cohort (N = 620)	Hyperfiltrating Cohort (N = 152)	*p* Value
Demographic features				
Sex (male/female), N (%)	518/254 (67.1/33.9)	412/208 (66.4/33.6)	106/46 (69.7/30.3)	0.433
Age, years	47.3 ± 8.9	49.5 ± 7.7	38.4 ± 8.3	**<0.001**
Active smokers, N (%)	122 (15.8)	98 (15.8)	24 (15.6)	0.149
Metabolic features				
BMI, Kg/m [2]	31.5 ± 6.2	31.6 ± 6.2	31.4 ± 6.5	0.652
Obesity (BMI ≥ 30), N (%)	412 (53.4)	334 (53.8)	78 (51.4)	0.628
Hypertension, N (%)	316 (40.9)	260 (41.9)	56 (36.7)	**<0.01**
Dyslipidaemia, N (%)	547 (70.8)	444 (71.6)	103 (68)	**0.033**
T2DM, N (%)	281 (36.3)	242 (39.0)	39 (25.6)	0.834
Impaired fasting glucose, N (%)	33 (4.3)	28 (4.5)	5 (3.3)	0.656
Previous CV events, N (%)	51 (6.6)	46 (7.4)	5 (3.6)	0.215
Therapy				
ACE-inhibitors/ARBs use, N (%)	227 (29.4)	198 (31.9)	29 (18.8)	0.281
Calcium antagonists use, N (%)	144 (18.6)	115 (18.5)	29 (18.8)	0.875
Beta-blockers use, N (%)	121 (15.6)	83 (13.4)	38 (25.0)	0.289
Diuretics use, N (%)	80 (10.4)	51 (8.2)	29 (18.8)	0.345
Statins use, N (%)	96 (12.4)	83 (13.4)	13 (8.3)	0.624
Hepatic features				
Steatosis grade, N (%)				**0.007**
1	252 (32.6)	211 (34)	41 (26.9)	
2	108 (13.9)	88 (14.2)	20 (13.1)	
3	228 (29.5)	168 (27.1)	60 (39.5)	
FibroScan stiffness value (kPa)	7.6 (2.2–52.2)	7.5± 6.1	8.0± 6.1	**0.023**
FibroScan logarithmic stiffness value (kPa)	0.81 (0.34–1.72)	0.79 ± 0.22	0.83 ± 0.22	**0.023**
FibroScan ≥ 7.2 kPa, N(%)	234 (30.3)	180 (29)	54 (35.5)	**<0.001**
FibroScan CAP measurement (dB/min)	316 ± 56	316 ± 56	320 ± 59	0.748
Kidney function				
Creatinine (micromol/L)	74.7 (37.0–125.0)	77.7 ± 14	63.5 ± 10.7	**<0.001**
eGFR MDRD study equation (mL/min/1.73 mq)	94.72 ± 21.3	89.1 ± 18.0	117.2± 6.5	**<0.001**
eGFR CKD-EPI 2021 (mL/min/1.73 mq)	96.6 ± 14.8	91.8 ± 11.9	117 ± 7.2	**<0.001**

Data expressed as mean ± standard deviation for normally distributed variables, and as median (1st–3rd quartile) for non-normally distributed variables. Numbers in bold represent statistical significance. Abbreviations: ACE, angiotensin converting enzyme; ARB, angiotensin receptor blockers; BMI, body mass index; CAP, controlled attenuation parameter; CV, cardiovascular; eGFR, estimated glomerular filtration rate; MDRD, modification of diet in renal disease; CKD-EPI, chronic kidney disease epidemiology collaboration.

**Table 2 ijms-24-15837-t002:** Binomial logistic regression for eGFR (CKD-EPI 2021) hyperfiltration by considering the covariate liver stiffness measurement as a continuous variable.

Variable	OR (95% CI)	*p* Value
Age, years	0.84 (0.82–0.87)	**<0.001**
Sex, male	1.56 (0.9–2.6)	0.110
T2DM	1.62 (0.9–2.9)	0.060
Hypertension	0.69 (0.90–1.22)	0.236
BMI ≥ 30	0.91 (0.57–1.46)	0.845
FibroScan stiffness logarithmic, kPa	6.6 (2.2–19.9)	**<0.001**

Numbers in bold represent statistical significance. Abbreviations: CKD-EPI, chronic kidney disease epidemiology collaboration; eGFR, estimated glomerular filtration rate; T2DM, type 2 diabetes mellitus.

**Table 3 ijms-24-15837-t003:** Binomial logistic regression for eGFR (CKD-EPI 2021) hyperfiltration considering hepatic fibrosis as a covariate as determined by liver stiffness measurement >7.2 kPa.

Variable	OR (95% CI)	*p* Value
Age, years	0.85 (0.82–0.87)	**<0.001**
Sex, male	1.48 (0.89–2.47)	0.125
T2DM	1.76 (0.98–3.13)	0.060
Hypertension	0.73 (0.42–1.28)	0.279
BMI ≥ 30	0.94 (0.59–1.51)	0.810
FibroScan stiffness > 7.2 kPa	1.83 (1.10–3.03)	**0.02**

Numbers in bold represent statistical significance. Abbreviations: CKD-EPI, chronic kidney disease epidemiology collaboration. eGFR, estimated glomerular filtration rate; T2DM, type 2 diabetes mellitus.

## Data Availability

The data are not published due to privacy restrictions. The authors could provide the dataset on request.

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
