# Peer review of "Glomerular Hyperfiltration: A Marker of Fibrosis Severity in Metabolic Associated Steatotic Liver Disease in an Adult Population"

_ijms, 2023, doi:10.3390/ijms242115837_

Round 1

Reviewer 1 Report

Comments and Suggestions for Authors

This manuscript evaluates the prevalence of GH in MASLD and its possible association with liver fibrosis. The results obtained may have implications for future screening of subjects with MASLD for abnormally elevated GFR values that may promote early recognition of possible risks for the development of CKD. Although the overall manuscript cannot further clarify the possible relationship between GH and fibrosis severity, the research design of the overall cohort study can generally meet the needs of the research purpose in terms of patient screening, clinical data collection, and statistical analysis. Partially revised and suitable for publication.

1. Since this study mainly uses Glomerular Hyperfiltration as a marker to predict the severity of fibrosis in metabolic-related fatty liver disease. Therefore, a literature description of Glomerular Hyperfiltration should be added to the preface of this manuscript to increase reader readability. For example, Glomerular Hyperfiltration has also been used to observe dementia, Sickle cell anemia-associated nephropathy, sleep quality, diabetes, obesity, adverse cardiovascular events, and even as an essential indicator of the likelihood of adverse health conditions (including malignant tumors).

2. Since the study is limited to the 40-65 age range, it is recommended to show this characteristic in the abstract or title.

3. In Table 1, why are there no data for women? And whether there are differences in gender factors in this study.

4. Please list the research ethics review approval document number certificate.

5. The numerical values and formulas used in OR (Odds Ratio) should be explained or listed.

6. The header of Table 3 should be distinguished from the header of Table 2.

Author Response

This manuscript evaluates the prevalence of GH in MASLD and its possible association with liver fibrosis. The results obtained may have implications for future screening of subjects with MASLD for abnormally elevated GFR values that may promote early recognition of possible risks for the development of CKD. Although the overall manuscript cannot further clarify the possible relationship between GH and fibrosis severity, the research design of the overall cohort study can generally meet the needs of the research purpose in terms of patient screening, clinical data collection, and statistical analysis. Partially revised and suitable for publication. 

1. Since this study mainly uses Glomerular Hyperfiltration as a marker to predict the severity of fibrosis in metabolic-related fatty liver disease. Therefore, a literature description of Glomerular Hyperfiltration should be added to the preface of this manuscript to increase reader readability. For example, Glomerular Hyperfiltration has also been used to observe dementia, Sickle cell anemia-associated nephropathy, sleep quality, diabetes, obesity, adverse cardiovascular events, and even as an essential indicator of the likelihood of adverse health conditions (including malignant tumors).

We thank the Reviewer for this comment. Indeed, we have already reported in the introduction the association between GH and metabolic alterations and CV risk (page 3, lines 76-79); now, we have also extended the description on the association between GH and other clinical conditions as dementia, cancers and sleep disorders as suggested: “ In addition, GH has been associated also to other pathological conditions as dementia, sleep disturbances and cancers, the latter both in the adult and infant population” (page 3, lines 80-81).

2. Since the study is limited to the 40-65 age range, it is recommended to show this characteristic in the abstract or title.

As reported in the method section, we enrolled patients with age 18-65 years, however by mistake in the flowchart it was reported age 40-65 ys. We have now corrected the age in the flowchart. Given the fact that the population we studied is made of adults, we also corrected the title as suggested into “Glomerular Hyperfiltration: a marker of fibrosis severity in Metabolic Associated Steatotic Liver Disease in an adult population”. We have specified it also in the text (page 4, line 97).

3. In Table 1, why are there no data for women? And whether there are differences in gender factors in this study.

We apologize to the Reviewer for not adding data on females in Table 1, now this information has been added (see in red in Table 1). In addition, the association between fibrosis assessed by Fibroscan and hyperfiltration was independent of sex, as reported in multivariate analysis in Tables 2 and 3. Finally, the prevalence of hyperfiltration did not differ between males and females in the whole cohort (20.5% vs 18.1%, p=0.43).

4. Please list the research ethics review approval document number certificate.

We apologize for not inserting the numbers in the text. We have now added the approval numbers for both Italian Centers (page 5, line 116).

5. The numerical values and formulas used in OR (Odds Ratio) should be explained or listed.

As for the Reviewer’s request, we have better specified the meaning of OR in the statistical analysis section (page 6, lines 166-167).

6. The header of Table 3 should be distinguished from the header of Table 2.

We really apologize for this oversight; we have now differentiated the two headings (in red).

Table 2. Binomial logistic regression for eGFR (CKD-EPI 2021) hyperfiltration by considering as covariate liver stiffness measurement as continuous variable.

Table 3. Binomial logistic regression for eGFR (CKD-EPI 2021) hyperfiltration by considering as covariate hepatic fibrosis by liver stiffness measurement >7.2 kPa.

Reviewer 2 Report

Comments and Suggestions for Authors

I was pleased to read this original article on a multi-centre collaboration led by Professor Lombardi. The identification of factors associated with the presence and risk of progression of liver fibrosis is a crucial point in the management of patients with MASLD. Glomerular hyperfiltration is objectively a novelty in this field and the authors well document its value and potential. 

The article is well written. The statistical analysis is simple and straightforward. I have only a few comments:

- Explain the use of CAP in the methods (it is included in the results, but is not defined in the methods)

- What do the authors mean by "pre-diabetes" (table 1)? It should be defined in the methods (just as the diagnosis of diabetes has been defined). Would it not be more appropriate to use the term 'Impaired Fasting Glucose'?

- I think the choice of the cut-off of 7.2 kPa for the indentification of liver fibrosis should be better justified/explained in the methods. The cited work (Foucher et al, Gut, 2006) identifies this value to identify patients with moderate fibrosis (⩾2): I think this should be clarified in the paper. 

Author Response

I was pleased to read this original article on a multi-centre collaboration led by Professor Lombardi. The identification of factors associated with the presence and risk of progression of liver fibrosis is a crucial point in the management of patients with MASLD. Glomerular hyperfiltration is objectively a novelty in this field and the authors well document its value and potential. 

The article is well written. The statistical analysis is simple and straightforward. I have only a few comments:

- Explain the use of CAP in the methods (it is included in the results, but is not defined in the methods)

We really apologize to the Reviewer for this oversight; we have now added information about CAP in both methods (page 6, lines 159-160) and results (page 7, line 183) sections.

- What do the authors mean by "pre-diabetes" (table 1)? It should be defined in the methods (just as the diagnosis of diabetes has been defined). Would it not be more appropriate to use the term 'Impaired Fasting Glucose'?

We completely agree with the Author, as we meant IFG. We have now amended the nomenclature both in the text (page 5, lines 129-130) and in Table 1.

- I think the choice of the cut-off of 7.2 kPa for the indentification of liver fibrosis should be better justified/explained in the methods. The cited work (Foucher et al, Gut, 2006) identifies this value to identify patients with moderate fibrosis (⩾2): I think this should be clarified in the paper. 

We really thank the Reviewer for this very interesting suggestion. We have better explained our choice of detecting significant fibrosis by LSM in the text “Given the middle age of our population and being GH an early marker of renal dysfunction, we decided to evaluate the presence of initial stages of hepatic fibrosis by detecting significant liver fibrosis by using a LSM >7.2 kPa according to international consensus” (page 6, lines 155-158).
